# An Active Learning Performance Model for Parallel Bayesian Calibration of Expensive Simulations

**Özge Sürer**
Information Systems & Analytics
Miami University
Oxford, OH 45056
surero@miamioh.edu

**Stefan M. Wild**
Applied Math & Computational Research Division
Lawrence Berkeley National Laboratory
Berkeley, CA 94720
wild@lbl.gov

## Abstract

Estimating parameters of simulation models based on observed data is an especially challenging task when the computational expense of the model – necessitated by faithfully capturing the real system – limits the learning process. When simulation models are expensive to evaluate, emulators are often built to efficiently approximate model outputs during model calibration. Computing-informed active learning, guided by intelligent acquisition functions, can improve data collection for emulators, thereby enhancing the calibration's efficiency. However, the performance of active learning strategies depends on computational factors such as computing environment (e.g., parallel resources available), tradeoffs in (calibration and simulation) algorithm's ability to benefit from parallelism, and the computational expense of the simulation models. In addition to overviewing these considerations, this work provides examples exemplifying the tradeoffs that make such learning difficult.

## 1   Introduction

Simulation models are widely used across engineering and science disciplines to explain complex systems. These models require calibration parameters as input to produce outputs that closely represent reality. However, calibration parameters are often unknown and must be inferred from data. The calibration process becomes particularly challenging when each simulation run is computationally expensive. Bayesian calibration (Sung and Tuo 2024), which quantifies uncertainty in both parameters and predictions, offers a solution by using emulators such as Gaussian processes (GPs) (Rasmussen and Williams 2005, Gramacy 2020) to approximate a simulation model's behavior. Emulators, often built from carefully selected a priori designs, help facilitate the calibration process (see Santner et al. (2018) for a thorough review). However, traditional a priori design methods may not adequately explore the input space, especially in higher-dimensional parameter settings, potentially limiting the emulator's accuracy for calibration. Therefore, careful selection of experimental designs for simulation evaluations is crucial for precise calibration with limited simulation runs.

This work explores active learning for sequential simulation data collection to build statistical emulators for calibration. In active learning, an acquisition function assesses the value of evaluating the simulation model at any given parameter. Parameters can be acquired sequentially, either one at a time (fully sequential) or in batches. While fully sequential procedures can yield more accurate, evaluation-efficient results given their use of all information obtained up to that point, batch sequential approaches may achieve high-quality solutions more quickly by evaluating multiple parameters simultaneously in parallel processing environments. However, the performance of active learning is influenced by various computational factors, including the convergence rate of the acquisition function (and its variation with batch size), the time complexity of the acquisition function with different batch

Workshop on Bayesian Decision-making and Uncertainty, 38th Conference on Neural Information Processing Systems (NeurIPS 2024).

and parallelism sizes, and the characteristics of the simulation model (e.g., its parameter dependence and run-time variability). In this work, we propose a performance model to understand the impact of these computational factors, ultimately guiding the selection of the best configuration in practice.

In § 2 we present an acquisition function for calibration, which is used to gather experimental data to motivate our experiments in § 4. § 3 introduces the performance model to investigate the effect of computational factors.

## 2    Acquisition Function for Calibration

We consider a simulation model $\eta(\cdot)$ that takes a parameter $\boldsymbol{\theta} \in \Theta \subset \mathbb{R}^p$ and returns an output $\eta(\boldsymbol{\theta})$, and a data generation mechanism yielding $y = \eta(\boldsymbol{\theta}) + \epsilon$, $\epsilon \sim \mathcal{N}(0, \sigma^2)$. At each stage indexed by $t$, the simulation data $\mathcal{D}_t = \{(\boldsymbol{\theta}_i, \eta(\boldsymbol{\theta}_i)) : i = 1, \dots, n_t\}$, collected sequentially, is stored to build a GP emulator. Using a GP emulator, $\eta(\boldsymbol{\theta})|\mathcal{D}_t$ at any $\boldsymbol{\theta}$ follows a normal distribution with mean $m_t(\boldsymbol{\theta})$ and variance $s_t^2(\boldsymbol{\theta})$.

Calibration can have multiple objectives and different acquisition functions are tailored to achieve specific goals. For simplicity, we focus only on enhancing parameter inference by learning the posterior density. To this end, we use the expected integrated variance criterion (EIVAR) (Sürer et al. 2024, Sürer 2024), which selects parameters that minimize the aggregated uncertainty of the posterior $p(\boldsymbol{\theta}|y)$. EIVAR at any $\boldsymbol{\theta}^*$ is computed as $\mathcal{A}(\boldsymbol{\theta}^*) = \int_{\Theta} \mathbb{E}_{\eta(\boldsymbol{\theta}^*)|\mathcal{D}_t} \left( \mathbb{V}\left[ p(\boldsymbol{\theta}|y) \right| (\boldsymbol{\theta}^*, \eta(\boldsymbol{\theta}^*)) \cup \mathcal{D}_t] \right) d\boldsymbol{\theta}$. The derivation of EIVAR follows from Sürer et al. (2024) and can be approximated over a set of (e.g., uniformly distributed) reference parameters $\Theta_{\mathrm{ref}}$ as

$$\mathcal{A}(\boldsymbol{\theta}^*) \approx \frac{1}{|\Theta_{\mathrm{ref}}|} \sum_{\boldsymbol{\theta} \in \Theta_{\mathrm{ref}}} p(\boldsymbol{\theta})^2 \left( \frac{f_{\mathcal{N}}\left( y;\, m_t(\boldsymbol{\theta}),\, \frac{1}{2}\left( \sigma^2 + s_t^2(\boldsymbol{\theta}) + \tau_t^2(\boldsymbol{\theta}, \boldsymbol{\theta}^*) \right) \right)}{2\pi^{1/2}|\sigma^2 + s_t^2(\boldsymbol{\theta}) - \tau_t^2(\boldsymbol{\theta}, \boldsymbol{\theta}^*)|^{1/2}} \right), \qquad (1)$$

where $\tau_t^2(\boldsymbol{\theta}, \boldsymbol{\theta}^*) = \mathrm{cov}_t(\boldsymbol{\theta}, \boldsymbol{\theta}^*)^2 / \left( s_t^2(\boldsymbol{\theta}^*) + \upsilon_t \right)$, $\mathrm{cov}_t(\boldsymbol{\theta}, \boldsymbol{\theta}^*) = k_t(\boldsymbol{\theta}, \boldsymbol{\theta}^*) - \mathbf{k}_t(\boldsymbol{\theta})^\top \mathbf{K}_t^{-1} \mathbf{k}_t(\boldsymbol{\theta}^*)$, and $f_{\mathcal{N}}$ evaluates a normal probability density. Here, $\upsilon_t > 0$ is the nugget parameter, $k_t(\cdot, \cdot)$ is the kernel function and $\mathbf{k}_t(\boldsymbol{\theta})$ consists of the cross-kernel evaluations between $\boldsymbol{\theta}$ and $\boldsymbol{\theta}_i$, for $i = 1, \dots, n_t$. The matrix $\mathbf{K}_t$ is an $n_t \times n_t$ matrix with $(i, j)$-th entry $k_t(\boldsymbol{\theta}_i, \boldsymbol{\theta}_j) + \upsilon_t \mathbb{1}_{i=j}$ for $1 \leq i, j \leq n_t$. Both (Sürer et al. 2024, Sürer 2024) provide a summary of the accuracy of EIVAR and other benchmark methods across different stages, operating under the assumption that the run time for each simulation evaluation is fixed and that acquisition times are negligible. In contrast, this work aims to illustrate how the performance of active learning methods is influenced by various computational factors, including variable run times and acquisition times, thereby providing a more comprehensive understanding of their effectiveness in a multitude of real-world scenarios.

We assume access to $w$ workers, which can run $w$ simulation models concurrently, and an additional worker that generates $b$ parameters with the acquisition function. If $w = 1$, we conduct data collection sequentially, one parameter at a time. For $w > 1$, updates are performed either synchronously or asynchronously, depending on the batch size $b$. When $b = 1$, we generate a set of candidate parameters from the prior and select the one that minimizes (1). For $b > 1$, rather than solving $b \times p$-dimensional optimization problem, we minimize (1) $b$ times over a candidate set of parameters. At each of the $b$ iterations, we assume a constant value for the unknown simulation output of the selected parameter and then update the emulator. This approach is known as the constant liar strategy in Bayesian optimization (Ginsbourger et al. 2010). In this context, the constant liar strategy serves to illustrate the proposed batch-sequential procedure. However, this strategy may introduce inaccuracies in predictions because it relies on fixed estimates for unseen parameters, which might not accurately represent the actual simulation outputs. Alternative approaches for batch acquisitions could be employed, each leading to different performance tradeoffs that affect both computational efficiency and the quality of the solutions obtained.

## 3    Performance Model Using Active Learning

Algorithm 1 presents the pseudocode for a performance model utilizing the sequential data collection described in § 2. The algorithm collects $n(b, \alpha)$ simulation data points to achieve $\alpha$-level calibration accuracy with batch size $b$ (line 4). To conduct our numerical analysis, we record the time $c_j^{\mathcal{J}}(b, w)$ at which each job $j$ ended, which is equal to the sum of the start time of job $j$ and the random execution

---

**Algorithm 1:** Performance model for parallel calibration

**1** **Input**: batch size $b$, worker size $w$, number of data to be collected $n(b, \alpha)$

**2** *Initialize* $t = 0$; $\mathcal{P}_t(w) = \{1, \ldots, w\}$; $n_t = w$

**3** $c_j^{\mathcal{J}}(b, w) = s_j \quad \forall j = 1, \ldots, w$; $c_t^{\mathcal{S}}(b, w) = 0$

**4** **while** $n_t < n(b, \alpha)$ **do**

**5**     $t \leftarrow t + 1$; $n_t \leftarrow n_{t-1}$; $\mathcal{P}_t(w) \leftarrow \mathcal{P}_{t-1}(w)$

**6**     $c_t^{\mathcal{S}}(b, w) = \max \left\{ c_{t-1}^{\mathcal{S}}(b, w), \left\{ c_j^{\mathcal{J}}(b, w) : j \in \mathcal{P}_t(w) \right\}_{[b]} \right\} + a(b, t)$

**7**     *Remove* $b$ completed jobs from pending $\mathcal{P}_t(w)$

**8**     **for** $i = 1, \ldots, b$ **do**

**9**         *Generate* job $n_t + i$

**10**         $c_{n_t+i}^{\mathcal{J}}(b, w) = c_t^{\mathcal{S}}(b, w) + s_{n_t+i}$

**11**         $\mathcal{P}_t(w) \leftarrow \mathcal{P}_t(w) \cup \{n_t + i\}$

**12**     $n_t \leftarrow n_t + b$

---

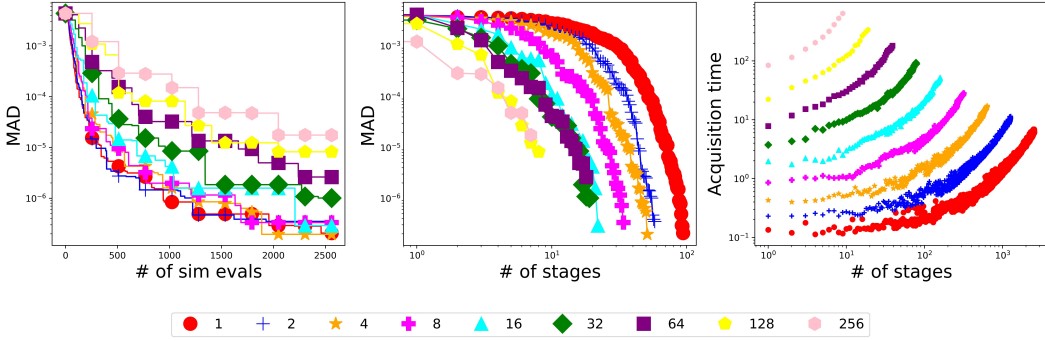

Figure 1: Calibration problem characteristics for EIVAR on a test function with $w = 256$.

time $s_j$ (line 10). The execution time varies across different simulation evaluations because different parameters can affect the computational complexity of the simulation model and may result in slower (or faster) convergence. Each job begins after the stage at which its input acquisition has completed.

We store the end time $c_t^{\mathcal{S}}(b, w)$ for each stage $t$ (line 6). The acquisition time $a(b, t)$ depends on the stages and batch sizes. In our case, since the time required to build a new emulator increases with the growing size of the simulation data, $a(b, t)$ increases in both batch size and time. The pending list $\mathcal{P}_t(w)$ maintains the indices of jobs submitted for evaluation that have not yet completed (line 11). This list helps track ongoing simulations, allowing the system to manage and monitor which tasks are still in progress. At each stage, the job in the $b$th order of the pending list is the last one for which the simulation output was received from the workers.

This performance model also allows one to employ problem characteristics (e.g., distributions of the number of simulations required, simulation run times, and acquisition times) to assess the wall time needed to obtain a specific accuracy, and thereby choose an acquisition and computing configuration.

## 4 Results and Discussion

We acquired 2560 parameters using EIVAR with the well-known Himmelblau function (Surjanovic and Bingham 2013), experimenting with batch sizes $b \in \{1, 2, 4, 8, 16, 32, 64, 128, 256\}$ and a worker size $w = 256$. Detailed information about the experimental setup is provided in Appendix A. The accuracy is assessed by the mean absolute deviation (MAD) between the true and estimated posteriors. In a sequential design, performance is typically summarized after receiving new evaluations, as illustrated in the left panel of Figure 1. The performance degrades with an increasing batch size because more parameters are chosen without incorporating new information from the evaluations of other parameters in the same batch. The performance summary across stages (middle panel) can

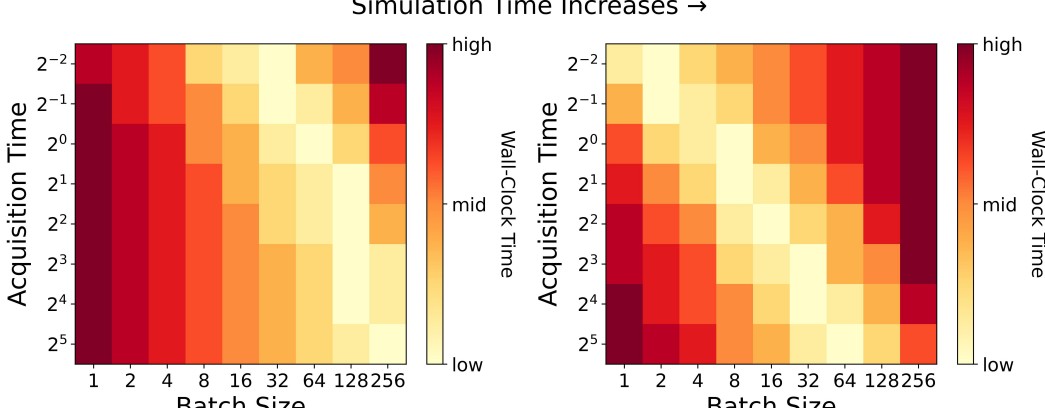

Figure 2: Wall-clock time to achieve an $\alpha$-level solution with various acquisition times and batch sizes. The mean simulation time increases from $\mu = 2^1$ in the left panel to $\mu = 2^6$ in the right panel with the simulation time standard deviation set to $\mu$. The $y$-axis shows different $\breve{a}$ values.

approximate the wall-clock time required for data collection, assuming that simulation run time is relatively constant and is the dominant computational cost. However, as acquisition times vary across stages and batch sizes (right panel), combining these factors with the computational expense of the simulation models may lead to significant variability in overall performance.

To investigate this, we use the progress curves in the left panel of Figure 1 to provide $n(b, \alpha)$ as input to the performance model in Algorithm 1. We evaluate this model under varying simulation and acquisition times. We model random simulation run times using a truncated normal distribution. We apply a linear increase to the acquisition time, varying the slopes and intercepts. For each job $j \in \{1 + b(t - 1) : t = 1, \ldots, n/b\}$, the linear acquisition time $\left(\breve{a} + \breve{b}j/n\right)$ with $\breve{a} = \breve{b}$ is obtained. The $\alpha$-level corresponds to the highest accuracy, defined as the lowest MAD, achieved with $b = 256$. For comparison, we determine $n(b, \alpha)$ for other batch sizes (e.g., $n(b = 1, \alpha) = 246$). Figure 2 illustrates the wall-clock time required to achieve an $\alpha$-level accuracy across different batch sizes and acquisition times. When acquisition times are relatively longer, larger batch sizes generally reduce the wall-clock time needed to achieve $\alpha$-level accuracy, as smaller batch sizes require more frequent acquisitions. While smaller batch sizes require fewer simulation evaluations, the wall-clock time increases due to the total acquisition time. The best batch size varies with different acquisition and simulation times. For instance, in the right panel, $b = 16$ minimizes wall-clock time when the acquisition time is $2^2$, whereas $b = 128$ is the best in the left panel for the same acquisition time.

A key benefit of our performance model is that it allows one to investigate various settings, including when the simulation run times are constant, have multiple modes, or exhibit other variability patterns. Figure 3 examines how variability in simulation run times influences the batch size for minimizing wall-clock time, considering different simulation and acquisition times. As variability increases (right panel), synchronous updates are beneficial only when acquisition times are significantly long. Smaller batch sizes perform better as the ratio of simulation times to acquisition time becomes larger. As a result, the best batch size achieves a balance between minimizing simulation evaluations and limiting acquisition overhead, depending on the relative scales of acquisition and simulation times.

## 5 Concluding Thoughts

What is the "best" active learning method? This paper shows that this depends on factors such as the computational environment (e.g., available parallelism, method scalability) and problem characteristics (e.g., simulation/acquisition times and their variability). A method that is best in one setting can readily be shown to perform poorly in another. Our proposed performance model is designed to help inform practitioners of the tradeoffs associated with candidate methods for their particular problem and setting. For instance, practitioners could test the candidate methods in a simulated environment using a similar but much less computationally expensive synthetic function

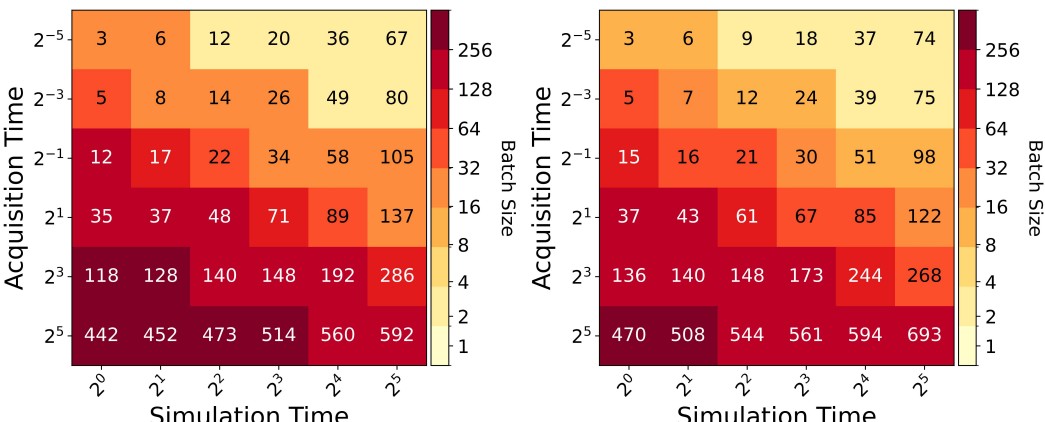

Figure 3: Batch size with various simulation and acquisition times. Colors indicate the batch size that minimizes the wall-clock time to achieve an $\alpha$-level solution and numbers in each cell represent the associated walk-clock time. The axes show the mean simulation time ($\mu$) and $\breve{a}$. The standard deviation in simulation time increases from $\mu \times 0.01$ in the left panel to $\mu$ in the right panel.

with similar characteristics to the target problem. They could also take empirical data on past simulation run times, apply knowledge on the scalability of the approach, and use all of these in our performance model in order to determine actions on the production (very computationally expensive) run. The aim in doing so is to maintain solution quality while being as efficient as possible with computational resources and wall time goals.

Future work could broaden the experimental evaluation by incorporating real-world simulations or more complex systems beyond the Himmelblau function. Testing on more intricate models or real calibration scenarios would enhance the robustness and generalizability of the findings, offering insight into the performance of active learning in a broad array of practical applications. Additionally, expanding the scope of acquisition strategies by exploring alternatives such as expected improvement or Thompson sampling, alongside the EIVAR criterion illustrated here, could provide a more thorough comparison. Such an exploration may reveal the potential advantages or limitations of each approach, improving our understanding of the effectiveness of each acquisition strategy across various calibration contexts. One limitation of the performance model is its reliance on the input number $n(b, \alpha)$ of data points to be collected. To address this, a predictive model could further be developed to estimate $n(b, \alpha)$ based on the characteristics of the simulation model, enhancing the performance model's usability and effectiveness. Many of these discussions are detailed in the extended paper by Sürer and Wild (2024).

## Acknowledgments and Disclosure of Funding

This work was supported in part by the NSF CSSI program under grant OAC-2004601 (BAND Collaboration) and by the U.S. Department of Energy, Office of Science, Office of Advanced Scientific Computing Research's SciDAC and applied mathematics programs under Contract No. DE-AC02-05CH11231. Computing resources provided on Bebop, a high-performance computing cluster operated by the Laboratory Computing Resource Center at Argonne National Laboratory, are gratefully acknowledged.

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

## A  Appendix / supplemental material

The details of the Himmelblau function are given as follows in the calibration context. We consider $\theta_1 \in [-5, 5]$ and $\theta_2 \in [-5, 5]$ and take $\boldsymbol{\theta} = (\theta_1, \theta_2)$ and $\eta(\boldsymbol{\theta}) = (\theta_1^2 + \theta_2 - 11)^2 + (\theta_1 + \theta_2^2 - 7)^2$ with $y = 1$ and $\sigma^2 = 1$. The function has a multimodal density as shown in Figure 4.

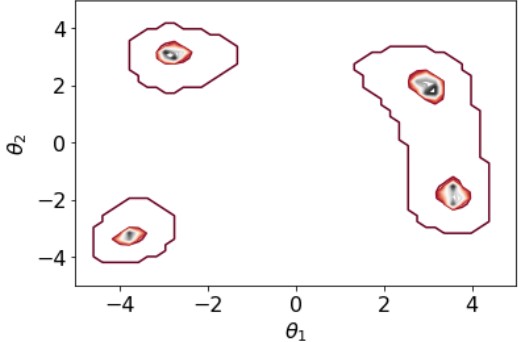

Figure 4: Illustration of the Himmelblau function's synthetic density.

