# OpenReview forum: "An Active Learning Performance Model for Parallel Bayesian Calibration of Expensive Simulations"
_NeurIPS.cc/2024/Workshop/BDU — NeurIPS BDU Workshop 2024 Poster_

### Official Review · Reviewer_T5gW · 2024-09-20
**Review for An Active Learning Performance Model for Parallel Bayesian Calibration of Expensive Simulation**

**Rating:** 6
**Confidence:** 3

**Review:**

1. Summary of the Paper
The paper proposes a performance model for active learning in the context of parallel Bayesian calibration of expensive simulations. It addresses the computational challenges in optimizing calibration efficiency using Gaussian Process (GP) emulators for high-cost simulation models. By introducing an acquisition function and a performance model, the authors analyze the trade-offs between batch size, parallelism, and acquisition time, demonstrating how these factors affect the performance of Bayesian calibration in practice.
2. Strengths
• The work tackles an important challenge in simulation-based learning where the cost of running simulations necessitates the use of efficient calibration techniques. The focus on parallelism and active learning makes the contribution particularly relevant in domains involving high-performance computing.
• The authors employ a well-structured set of experiments, including variations in batch sizes and worker configurations, to illustrate the effect of computational factors on calibration performance. The use of both synthetic functions (Himmelblau function) and performance metrics (mean absolute deviation) effectively validates the proposed model.
• The paper is well-organized, with a clear progression from introducing the calibration problem, formulating the acquisition function, and finally presenting the performance model and experiments. This logical structure aids the reader in understanding the flow of ideas.
3. Weaknesses
• While the empirical findings are compelling, the paper lacks deeper theoretical justifications for some of the observed phenomena. For instance, the impact of batch size on the efficiency of active learning is primarily discussed from an empirical perspective. A more rigorous theoretical exploration would strengthen the contribution.
• The work relies on a simulated function (Himmelblau) for the bulk of the experimental evaluation. Though effective for illustrating basic principles, it does not fully capture the complexity of real-world calibration tasks. Incorporating results from real-world simulations or more complex systems would make the findings more generalizable.
• The paper focuses solely on the EIVAR criterion for acquisition. While it is a reasonable choice, the inclusion of alternative acquisition functions, such as expected improvement or Thompson sampling, could provide a more comprehensive comparison and reveal potential advantages or limitations of the selected criterion.
4. Minor Comments
• Typographical errors are minimal, but a final proofreading pass is recommended.
• The paper would benefit from a stronger conclusion that ties together the empirical findings and provides more concrete recommendations for practitioners.

---

### Official Review · Reviewer_s9t5 · 2024-10-03
**Peer review for An Active Learning Performance Model for Parallel Bayesian Calibration of Expensive Simulations**

**Rating:** 5
**Confidence:** 3

**Review:**

## Summary
This paper addresses the resource challenge of calibrating simulation models. The authors propose a performance model for active learning in parallel Bayesian calibration. By using Gaussian process emulators and informed acquisition functions, they aim to optimize data collection and improve the efficiency of the calibration process. The paper discusses how computational factors influence the performance of active learning strategies and provides empirical results illustrating the trade-offs involved in selecting batch sizes and acquisition times.

## Big-picture comments
* **Originality**: the paper introduces the performance model to investigate the effect of computation factors, which is an original contribution. However, the paper might benefit from a further discussion on how this approach addresses the gap in prior literature, such as Sürer et al. 2024
* **Study set-up**: the technical execution of this paper as well as the proposed acquisition function and performance model are clearly defined. It might be beneficial to discuss the limitations of the set-up (e.g., constant liar strategy in Bayesian optimization setting is itself a tradeoff between computational efficiency and performance).
* **Result Discussion / Implication**: while the results of the experiment are well-documented and visualized, the paper could benefit from a deeper discussion on the real-world implications of this study and the directions that future research should take.
* **Significance**: the research aims to address a significant problem facing researchers and practitioners in simulations, hence the research carriers real-world implications.

## Questions
1. Is the assumption of constant simulation run times in the performance model justifiable (line 95)?
2. How does the proposed approach compare with other active learning strategies in terms of performance and computational efficiency?
3. What are the specific implications of the proposed performance models? Are there any limitations in this approach?

---

### Decision · Program_Chairs · 2024-10-09

Accept (Poster)